# Can Resistance Exercise Be a Tool for Healthy Aging in Post-Menopausal Women with Type 1 Diabetes?

**DOI:** 10.3390/ijerph18168716

**Published:** 2021-08-18

**Authors:** Zeinab Momeni, Jessica E. Logan, Ronald J. Sigal, Jane E. Yardley

**Affiliations:** 1Physical Activity and Diabetes Laboratory, Li Ka Shing Centre for Health Research Innovation, Alberta Diabetes Institute, Edmonton, AB T6G 2E1, Canada; zmomeni@ualberta.ca (Z.M.); jelogan@ualberta.ca (J.E.L.); 2Augustana Faculty, University of Alberta, Camrose, AB T4V 2R3, Canada; 3Faculty of Kinesiology, Sport and Recreation, University of Alberta, Edmonton, AB T6G 2H9, Canada; 4Departments of Medicine, Cardiac Sciences and Community Health Sciences, Faculties of Medicine and Kinesiology, University of Calgary, Calgary, AB T2N 1N4, Canada; rsigal@ucalgary.ca; 5Women’s and Children’s Health Research Institute, Edmonton, AB T6G 1C9, Canada

**Keywords:** exercise, physical activity, resistance training, menopause, women, type 1 diabetes

## Abstract

Due to improvements in diabetes care, people with type 1 diabetes (T1D) are living longer. Studies show that post-menopausal T1D women have a substantially elevated cardiovascular risk compared to those without T1D. As T1D may also accelerate age-related bone and muscle loss, the risk of frailty may be considerable for T1D women. Exercise and physical activity may be optimal preventative therapies to maintain health and prevent complications in this population: They are associated with improvements in, or maintenance of, cardiovascular health, bone mineral density, and muscle mass in older adults. Resistance exercise, in particular, may provide important protection against age-related frailty, due to its specific effects on bone and muscle. Fear of hypoglycemia can be a barrier to exercise in those with T1D, and resistance exercise may cause less hypoglycemia than aerobic exercise. There are currently no exercise studies involving older, post-menopausal women with T1D. As such, it is unknown whether current guidelines for insulin adjustment/carbohydrate intake for activity are appropriate for this population. This review focuses on existing knowledge about exercise in older adults and considers potential future directions around resistance exercise as a therapeutic intervention for post-menopausal T1D women.

## 1. Introduction

Menopause can be a difficult transition for women, as it can have an impact on several aspects of physical health, in addition to affecting the quality of life. Several studies show that exercise and physical activity may help manage many of the physical symptoms experienced during and after menopause [1,2,3]. In addition, being active is known to improve functional fitness (the ability to perform daily activities of living) and ameliorate the quality of life [4,5]. Women with type 1 diabetes (T1D) may experience worse health outcomes with respect to cardiometabolic [6] and musculoskeletal health [7,8] with menopause than women without diabetes; however, they also have greater barriers to exercise due to their condition, such as fear of hypoglycemia [9]. Resistance exercise may be a promising approach for this population due to its specific impact on musculoskeletal [10,11] and cardiometabolic health [12,13] as well as its protection against age-related frailty [14]. In addition, resistance exercise may cause less decline in blood glucose during exercise than aerobic exercise [15]. There are currently no exercise studies involving post-menopausal women with T1D. This review discusses how exercise, and in particular resistance exercise, may be able to improve the physical and mental well-being of women with T1D as they age, and demonstrates the need for more research in this area.

## 2. Menopause and its Impact on Women’s Physical Well-Being

Menopause is the permanent cessation of menstruation, typically occurring naturally in a woman’s late 40s or early 50s. It is preceded by a 2–10-year period of perimenopause, during which the ovaries gradually produce less estrogen and progesterone. Menopause occurs when the ovaries stop releasing eggs [16]. These decreases in estrogen and progesterone levels are often associated with a variety of symptoms and conditions [17], the incidence and severity of which are highly variable [18]. In addition to the commonly identified hot flushes experienced by most women, musculoskeletal, metabolic, and cardiovascular complications, among others, have been widely reported after menopause.

A decline in muscle mass and strength, also known as sarcopenia, often occurs along with, and may be partly caused by, the decrease in estrogen that characterizes menopause [19]. In addition, low physical activity and age-related increases in oxidative stress and inflammation are among the greatest contributing factors for sarcopenia in post-menopausal women [19]. Menopause is also a critical period of change in bone mass and strength, which sets the stage for the development of osteopenia and osteoporosis, along with increased susceptibility to fractures [20]. Declines in bone mineral density (BMD) and a rapid phase of bone loss over the menopause transition [21] are well documented. The high prevalence of these musculoskeletal complications in post-menopausal women leads to higher incidences of falls and fractures, frailty, and subsequent morbidity and mortality in this population [8]. It needs to be noted that in addition to menopause, hyperglycemia-induced oxidative stress and accumulation of reactive oxygen species and advanced glycation end products also play roles in bone fragility by compromising bone collagen mineralization and, ultimately, bone strength, increasing marrow adiposity, and releasing inflammatory factors and adipokines from visceral fat which can potentially alter the function of osteocytes [22].

The prevalence of metabolic syndrome also increases with menopause [23]. The metabolic syndrome refers to the co-occurrence of several interconnected factors such as insulin resistance, obesity, atherogenic dyslipidemia, hypertension, and endothelial dysfunction, which increase the risk of developing cardiovascular disease (CVD) [24,25]. Menopause is often associated with changes in weight and body composition such as an increase in visceral abdominal fat deposition [26]. Alterations in lipid levels [27], such as increases in triglycerides and low-density lipoprotein cholesterol (LDL-C) and a decrease in high-density lipoprotein cholesterol (HDL-C), are among other CVD risk factors associated with menopause [23]. Increased risk of insulin resistance [23] and type 2 diabetes [28] as well as hypertension [29] have also been linked to menopause. These metabolic changes that emerge with estrogen deficiency after menopause may explain some of the elevated CVD risks in post-menopausal women [23].

## 3. Role of Exercise in the Management of Menopausal Symptoms

The pervasive burden of menopausal symptoms and complications across a wide array of health outcomes can have significant impacts on women’s quality of life [30,31,32]. As such, the management of symptoms in this population is essential. Regular exercise and/or physical activity is a safe, non-pharmacological approach to the management of several of these symptoms as it has been shown to decrease or alleviate many of them [3,33,34]. Table 1 provides a detailed summary of the studies examining the effects of physical activity and exercise on the management of musculoskeletal and cardiometabolic symptoms and quality of life in post-menopausal women.

### 3.1. Musculoskeletal Effects of Exercise

While the menopausal transition is associated with decreases in muscle mass at multiple anatomical levels, habitual participation in physical activity can maintain skeletal muscle mass during this transition [35]. For example, one moderate-to-vigorous intensity aerobic exercise program was able to preserve appendicular lean mass and skeletal muscle index (the ratio of skeletal muscle mass to height), despite no change in total lean mass in post-menopausal women [38]. The inclusion of resistance exercise, however, may be essential in increasing, rather than just preserving muscle mass and strength in this population. A training program consisting of both resistance and aerobic exercise significantly increased total body and regional lean soft tissue mass and decreased percentage of body fat in post-menopausal women [43]. A systematic review and meta-analysis, however, demonstrated that muscle strength and muscle function can be improved more than muscle mass by exercise programs such as aerobic training and resistance training in older adults with sarcopenia [50]. Resistance training improves neuromuscular adaptations including increased muscle strength [50], which would be particularly beneficial for post-menopausal women who experience a significant age-related decline in muscle force [51].

Resistance exercise, and in particular high-intensity resistance exercise, can also be an effective method to help prevent and reduce the severity of osteopenia and osteoporosis in aging women. Women are at a significantly higher risk of developing osteopenia and osteoporosis than men, as they have lower peak bone density, along with an earlier onset and faster rate of bone loss [52]. High-intensity resistance and impact training significantly improves bone density, functional performance relevant to falls, and decreases markers of frailty, while increasing lumbar spine and femoral head BMD in post-menopausal women [47].

The volume of exercise performed may also be a key factor in the impact of activity on bone density. While no difference was found in bone mineral content, one study found significantly higher BMD after a year in the post-menopausal women assigned to a higher dose of aerobic exercise as compared to those in a lower dose group [40]. It should be noted that the “low-dose” group was actually performing the recommended 150 min per week of moderate activity, indicating that post-menopausal women may, in fact, require more exercise than what is currently recommended in order to increase BMD.

While many studies vary in the exercise protocol being tested, a review of 43 randomized control trials examining exercise impacts on bone density in post-menopausal women found that the most effective intervention for improved BMD in the spine was a combined resistance and aerobic training program, while the most effective intervention for hip and femur BMD was resistance training [53]. It is important to note, however, that while the relationship between exercise and increased BMD is well established, the link between exercise and maintained whole bone strength is less clear [54]. Existing evidence around this topic relies mostly on a combination of exercise and hormone replacement therapy (HRT) [55], or nutritional supplements [56]. In general, these studies show that the combined effect of exercise and the added intervention (i.e., HRT and/or vitamin D) improve bone density in post-menopausal women and may improve whole bone strength immediately after intervention [55,56]. However, more research is needed on how exercise alone can influence menopause-induced decline in bone strength or metabolically-induced bone fragility, and whether improvements are maintained beyond the intervention period.

### 3.2. Metabolic and Cardiovascular Effects of Exercise

Regular exercise programs can be used as a means of weight management in post-menopausal women. For example, implementation of a moderate-intensity Nordic walking program resulted in reductions in total body fat, waist circumference, and body mass index (BMI) in pre- to post-menopausal women [37]. Similarly, another intervention study examining the effects of an endurance exercise program on central and abdominal adiposity in peri- and post-menopausal women showed a significant reduction in the waist-to-hip ratio in this population, without having an impact on BMI [45]. A yearlong moderate-to-vigorous intensity aerobic exercise intervention also produced significant reductions in overall and abdominal adiposity in post-menopausal women, with greater decreases among those with a higher exercise duration [39].

Resistance exercise is also beneficial for weight and adiposity management in this population. In a randomized clinical trial involving sedentary post-menopausal women, a long-term resistance exercise program led to significant weight and body fat losses in this population, especially among those with a higher exercise volume and frequency [49]. Similarly, in addition to a reduction in body fat percentage, one study showed that resistance exercise can decrease the metabolic syndrome severity Z-score with a concomitant lowering of fasting blood glucose and improvement in lean muscle mass in post-menopausal women [46]. The metabolic syndrome severity Z-score is a composite index of the severity of metabolic syndrome, taking into account the contributions of each component of the metabolic syndrome [57].

In addition to improvements in body composition, regular exercise and physical activity are beneficial for lowering other cardiovascular and metabolic risk factors. In an observational study during 458,018 woman-years of follow-up, walking, and total physical activity scores (based on weekly energy expenditure calculated in metabolic equivalents (MET)) were negatively correlated with risk factors for type 2 diabetes, especially BMI, in Caucasian post-menopausal women [58]. Similarly, while diabetes incidence was positively associated with BMI and the waist-to-hip ratio, it was negatively correlated with the frequency of both moderate and vigorous physical activity (self-reported and based on MET) in a cohort of 34,257 post-menopausal women [59]. Likewise, in a cross-sectional study, a lower risk of type 2 diabetes and a more favorable cardiovascular profile were found with higher levels of habitual physical activity (assessed by a digital pedometer), specifically walking, in a population of 292 middle-aged women, regardless of the menopausal status [60].

In line with the above evidence, a moderate-intensity physical training program followed by home-based physical activity targeting all major muscle groups reduced systolic and diastolic blood pressure, in addition to reducing BMI, waist-to-hip ratio, and LDL in sedentary post-menopausal women [36]. Similarly, resistance training led to significant reductions in total cholesterol, LDL-C, and non-HDL-C in this population [44]. Furthermore, a study of progressive resistance training showed that this activity could decrease weight, waist circumference, total cholesterol, LDL-C, and C-reactive protein in post-menopausal women, supporting the anti-inflammatory and the cardiometabolic benefits of exercise and physical activity in this population [48].

Overall, resistance exercise seems to be of particular importance in post-menopausal women as it can increase muscle mass and strength, and hip and femur BMD. This is of high importance due to the loss of skeletal muscle mass and strength with aging, and increased risk of hip injuries and frailty after the menopausal transition. In addition, through affecting the metabolic syndrome risk factors and improving lipid profile [61,62], resistance exercise can be considered an optimal strategy for preventing CVD and subsequent morbidity and mortality in this population.

### 3.3. Effects of Exercise on the Quality of Life

Menopausal and post-menopausal women who are regularly active have higher health-related quality of life scores than their sedentary counterparts [63]. An exercise intervention consisting of combined resistance and aerobic training found significant improvements in all six markers of quality of life (physical mobility, pain, sleep, energy, social isolation, emotional status) in post-menopausal women [41]. Longer-term interventions have produced similar results, with significant improvements in health-related quality of life in rural post-menopausal women who underwent a year-long customized exercise program of combined resistance, aerobic, flexibility, and relaxation exercises [42]. In addition to enhancing the quality of life directly, resistance training can also allow for a better quality of life indirectly, through promoting beneficial effects on muscles, bone, and adipose tissues in this population, as discussed earlier [64].

## 4. Type 1 Diabetes

Type 1 diabetes (T1D) is an auto-immune disorder in which the beta cells of the pancreas are destroyed, resulting in chronic insulin deficiency [65]. The absence or near-absence of endogenous insulin leads to hyperglycemia (high blood glucose), which must be treated by exogenous insulin, either through injections or an insulin pump. It is challenging to maintain continuously the balance of carbohydrate intake, physical activity, and exogenous insulin, and hypoglycemia (low blood glucose) often occurs [65]. People with T1D share many of the same benefits from exercise as their non-diabetic counterparts [66], in addition to the exercise-specific effects in this population such as reduced insulin requirements, reduced insulin resistance, and favorable changes in lipids [66].

### 4.1. Menopause in Women with T1D

As discussed earlier, menopause is associated with a wide range of symptoms across a wide array of health complications among women without diabetes. Throughout the life course, women with T1D tend to experience more complications related to the menstrual cycle and its cessation, many of which have negative consequences for cardiovascular and overall health. For example, compared to those without diabetes, women with T1D often experience delayed menarche and irregular menstrual cycles [67,68,69], which have been associated with increased coronary artery calcification (CAC) [70] and increased risk of fatal and non-fatal coronary heart disease (CHD) [71]. Because of such pre-existing conditions, it is reasonable to suspect that menopause would lead to more severe health consequences in women with T1D. There is, however, a need for a great deal more research in this understudied area.

Although T1D per se may not affect the age of onset of menopause [72,73], women with more severe microvascular complications of diabetes (such as retinopathy, neuropathy, and nephropathy [74]) are at greater risk of earlier menopause compared to other women with T1D and their non-diabetic counterparts [72,75]. Of note, lower age of menopause has been correlated with a higher risk of CVD and mortality [76,77]. This premise is also supported by data from a longitudinal study (*n* = 636) investigating the association between the menopausal transition and subclinical atherosclerosis in women with T1D, where higher CAC volume was found in this group as compared with non-diabetic women [6]. Moreover, differences in CAC volume between those with and without diabetes increased as women transitioned through menopause [6].

Compared to women without diabetes, those with T1D have greater excess risks of all-cause mortality, along with more fatal and non-fatal vascular events. The increase in risk with aging in women with T1D compared to women without diabetes is greater than the increase in risk experienced by T1D men compared to men without diabetes [78]. In particular, females with T1D are generally more insulin resistant [79], have more unfavorable changes in their fat distribution [80], and tend to develop a more atherogenic lipoprotein profile [81] with aging as compared to males with T1D. These metabolic risk factors, independently or together, put females with T1D at a significantly higher risk of developing CVD than their non-diabetic counterparts [82].

In addition to CVD, a significantly high risk of fractures is also reported in women with T1D. In a large (*n* = 334,266) population-based cohort study, a higher risk of fractures was reported in individuals with T1D compared to those without diabetes, particularly after the age of 40. The risk of hip fractures was greatest in the 80- to 90-year age bracket for both sexes, at 244.5 and 116.1 fractures per 10,000 person-years in women with and without T1D, and 76.7 and 59.6 fractures per 10,000 person-years in men with and without T1D, respectively [83]. Similarly, in another observational study, post-menopausal women with T1D were at least 12 times more likely to report an incident hip fracture than their non-diabetic counterparts [84]. In line with this report, a 15-year longitudinal study (*n* = 10,981) showed that women with T1D had more falls, incident fractures, and osteoporosis as compared to non-diabetic women across the menopausal transition [85], which could be attributed to the lower BMD [86] or lower bone quality [87] in this population. Moreover, many of the menopausal conditions discussed above have been shown to negatively impact quality of life [63,88,89], although there is insufficient research directly considering the interaction between T1D and menopause on quality of life. Further research is, therefore, warranted in this area.

With the majority of research in the field focusing on type 2 diabetes, there is limited research on menopause in women with T1D. Further research is strongly needed to determine how T1D affects the presence, severity, and management of menopausal symptoms in this high-risk group. Given the importance of exercise and physical activity in the management of menopausal symptoms in women without T1D, it is reasonable to consider physical activity and exercise as practical strategies for the management of menopausal symptoms in women with T1D.

### 4.2. Exercise and T1D

Regular exercise (at least 150 min per week) is recommended in both adults with and without T1D to maintain a balanced and healthy lifestyle [90,91], by improving cardiorespiratory fitness, muscular strength [92], mental health [93], and quality of life [94]. In addition, exercise also lowers the risk of a variety of chronic conditions, such as type 2 diabetes [93], CVD [95], hypertension [96], and dementia [97], while slowing age-related decline in physical function [98]. In those with T1D in particular, exercise and physical activity are associated with not only greater longevity [99,100], but also a decrease in the frequency and severity of diabetes-related complications [101,102,103].

People with T1D who exercise regularly have lower all-cause (hazard ratio 0.66) [104] and cardiovascular mortality. One large (*n* = 2639) longitudinal study of people with T1D showed that the 10-year cumulative cardiovascular mortality rates were 4.7% in low (<10 MET-h/week), 1.9% in moderate (10–40 MET-h/week), and 1.8% in high (>40 MET-h/week) leisure-time physical activity participants, respectively. In addition, increased frequency of physical activity was associated with a lower risk of cardiovascular mortality, with rates of 5.5% in low (fewer than one session/week), 2.8% in moderate (1–2 session/week), and 2.2% in high (more than 2 sessions/week) exercise frequency groups [105].

Where CVD is concerned, a cross-sectional study on males and females with (*n* = 105) and without (*n* = 176) T1D (mean age 39 ± 14 vs. 38 ± 12 years, respectively) found that three or more episodes of self-reported vigorous physical activity per week were associated with reduced CVD risk through the preservation of small artery compliance, independent of age, sex, and diabetes status [106]. Greater large artery compliance and pulse rate, however, were significantly associated with the frequency of physical activity only in the T1D group [106]. In addition, a prospective cohort study (*n* = 2185) of males and females (mean age 32.7 ± 10.2 years) with T1D found an inverse association between self-reported baseline physical activity and all-cause mortality in both sexes. Incident CVD, however, was inversely correlated with baseline physical activity only in women in the longitudinal analysis (*n* = 1063). Both walking distance and total physical activity were inversely associated with prevalent CVD in both sexes in the cross-sectional analysis (*n* = 1690) [104].

Similarly, in a cross-sectional study of 18,028 adults (mean age 33.8 ± 7.5 years) with T1D, an inverse relationship was found between self-reported physical activity and several CVD factors, including BMI, hypertension, and dyslipidemia [107]. Self-reported physical activity was also negatively correlated with hemoglobin A1c (HbA1c), diabetic ketoacidosis, microalbuminuria, and retinopathy in this population [107]. Another cross-sectional study of 1945 males and females (mean age 38.5 ± 12.3 years) with T1D showed less leisure-time physical activity as well as low-frequency and low-intensity leisure-time physical activity in those with diabetic nephropathy and proliferative retinopathy than in those without these complications [108]. In particular, low-frequency (one session/week) and low-intensity physical activity were associated with diabetic nephropathy, while low-intensity physical activity was associated with proliferative retinopathy and CVD in this cohort of T1D participants [108]. It should be noted that although these findings suggest the beneficial role of higher frequency and/or intensity of physical activity in the management of diabetes complications, the hindering impact of these chronic disabling complications on physical activity level in this population should not be overlooked.

Exercise and physical activity are also associated with more favorable body composition, BMI, BMD, and osteopenia in those with T1D. A cross-sectional study on 75 males and females with T1D (mean age 43.5 ± 10.5 years) and 75 counterparts without diabetes (mean age 40.1 ± 12.8 years) showed that having an active lifestyle (physical activity level ≥ 1.7) was associated with a lower BMI, a lower total and truncal fat mass, as well as a lower waist circumference as compared to those with a more sedentary lifestyle [109]. Similarly, in an intervention study involving 24 males and females with T1D with osteopenia (mean age 17.1 ± 2) and 38 control individuals without diabetes (mean age 16.9 ± 1.8), a three-month aerobic exercise program (on ergometer with constant speed and resistance, 70 min including warm-up and rest, 3 times/week) in the T1D group significantly increased BMD and serum procollagen type 1 N-terminal propeptide, reflecting improved bone formation [110].

Being physically active, however, can be challenging for those with T1D. At the onset of moderate-intensity aerobic exercise, uptake of glucose into active muscle cells increases. In people without diabetes, insulin secretion decreases, resulting in an increase in the glucagon:insulin ratio and increased hepatic glucose production, precisely matching the increased glucose utilization by muscles [111]. Although the same increase in muscle glucose uptake occurs in people with T1D, insulin levels are not regulated endogenously, and glucagon secretion is often impaired [112], so the glucagon:insulin ratio cannot increase. This imbalance leads to an insufficient increase in hepatic glucose production to match the increased glucose uptake into the skeletal muscle, subsequently inducing hypoglycemia, particularly during a longer duration of exercise [113]. These hypoglycemic episodes present a considerable barrier to exercise for people with T1D [9].

Catecholamines also play an important role in glucoregulation during exercise, particularly in very intense exercise. The onset of exercise triggers a release of catecholamines [114], which increases in proportion to the intensity and duration of exercise and in turn increases hepatic glucose production. In people with T1D, the increased hepatic glucose production can help override the insufficient change in glucagon:insulin ratio, thus reducing the risk of hypoglycemia during activity. The large catecholamine-induced increase in glucose production during brief, near-maximal intensity exercise can lead to hyperglycemia [115]. In those without T1D, insulin levels rise in response to the excess glucose and return blood glucose to normal. Without an increase in exogenous insulin, however, hyperglycemia can persist for hours after brief, intense exercise in people with T1D [116].

#### 4.2.1. Sex-Related Differences in Response to Exercise

In addition to the differences in response to exercise between those with and without T1D, there may also be sex-related differences in response to exercise. For example, females and males use similar fuel sources for energy at rest [117]. However, during prolonged submaximal exercise, females rely more on lipolysis of myocellular triacylglycerol than males, who rely more on glycogen stores [118]. Furthermore, males transition to using carbohydrates as their main fuel source earlier in anaerobic exercise than females [119]. The lower reliance on carbohydrates for fuel in females is generally reflected in less depletion of hepatic and muscle glycogen after exercise, although this can vary according to the type and duration of exercise, and stage in females’ menstrual cycles [120]. Following 60–90 min of isoenergetic moderate and hard-intensity exercise, women maintain blood glucose in a much tighter range than men, likely because of less depleted glycogen stores during exercise [121]. It should be noted, however, that most studies finding differences in fuel selection between males and females were focused on exercise in the fasting state [120].

There are also several hormone-related differences during exercise between females and males. Males tend to have a greater catecholamine response to similar levels of exercise [120]. Prior to menopause, females have a much higher level of estrogen, which contributes to the lower respiratory exchange ratio (i.e., greater reliance on lipids as a fuel source) during exercise [120]. Taken together, these hormonal differences create a reliance on different fuel sources between male and female participants [120]. While some of these differences are well-documented in individuals without T1D, exercise literature involving T1D participants is currently dominated by young fit males, bringing into question whether female participants may experience very different glucose trajectories during various types of exercise [122].

#### 4.2.2. Age-Related Differences in Response to Exercise 

In addition to these sex-specific differences, age also has an impact on the effects of, or responses to, exercise. Body composition, hormonal responses, cardiorespiratory fitness, and functional fitness are all affected by aging [123]. Aging causes changes in body composition such as decreases in lean body mass and bone density and increases in body fat and fat redistribution [124]. Hormonal changes such as decreased catecholamine response, and altered responses of growth hormone, cortisol, and glucagon also occur with aging [123]. In addition, aging causes reduced cardiorespiratory fitness, which is related to declines in peak oxygen uptake (VO_2_ peak) [125]. Lastly, declines in functional fitness, due to increased body fat percentage, loss of muscle mass in the extremities, as well as loss of flexibility, agility, and endurance also occurs with aging [126], playing potential roles in the effects of exercise in aging adults. In adults with T1D, both aging and longer diabetes duration are associated with greater insulin requirements [127,128], which can make the creation of safety recommendations around excise more challenging in this population. To date, however, there has been a complete lack of studies involving older participants with T1D.

It appears that sex, age, and the presence or absence of T1D all play roles in the effects of, and/or responses to, exercise and physical activity. These physiological factors place post-menopausal women with T1D in a unique position, where the interplay between sex, age, menopausal symptoms, and T1D complications is concerned. Nonetheless, there are currently no exercise studies involving this high-risk population, and as such, their acute blood glucose responses to exercise, as well as their response to longer-term training are essentially unknown. It can be suggested, however, that due to the many benefits of exercise in post-menopausal women it should be prescribed in the management of symptoms in post-menopausal women with T1D. More research is needed in order to ensure that exercise, and in particular resistance exercise, can be used as a therapeutic intervention without compromising blood glucose management in this population.

## 5. Exercise in Post-Menopausal Women with T1D

Unfortunately, most of the studies examining the effects of exercise and physical activity on T1D complications were conducted with both male and female participants and there is, therefore, a lack of studies with only female participants. Sex-related differences in counter-regulatory hormonal responses such as catecholamines and growth hormone to exercise exist among those without diabetes [120]. This includes a greater catecholamine response to various types of exercise in males, and a different pattern of growth hormone release during exercise such as a more prolonged response in males as compared to a higher, but transient, response in females [120]. In addition to the hormonal differences, being female, by itself, is associated with lower odds of achieving recommended physical activity levels (≥5 days/week) [129]. However, the impact that these factors may have on blood glucose responses to exercise in people with T1D is unclear. 

A recent secondary analysis highlighted potential sex-related differences in blood glucose responses to exercise. It examined responses to a program including 7 resistance-based exercises (3 sets of 8 repetition maximum (RM), ~45 min) in individuals with T1D [122] and found that female participants (mean age 29 ± 8 years) on average did not experience a decline in blood glucose during and after the resistance exercise session compared to male participants (mean age 34 ± 15 years) who experienced significant declines in glycemia. On the other hand, more female participants experienced post-exercise hyperglycemia than males in this secondary analysis [122]. Whether these observations reflect physiological differences in exercise responses or differences in behaviors related to diabetes management around exercise is currently unclear.

Despite such differences, current safety recommendations around exercise for individuals with T1D were developed using evidence from studies involving very few or no female participants, and almost uniquely younger individuals with T1D [123]. As such, the recommendations might not be appropriate for older females with T1D, such as those in the menopausal transition and post-menopause. Following these recommendations which were not developed or tested in studies of older T1D women might result in an increased risk of hypoglycemia, hyperglycemia, or greater fluctuations in blood glucose levels around exercise. Frequent hypoglycemia increases the risk of seizures and loss of consciousness [130], cardiac repolarization [131,132], all-cause mortality, and CVD [133]. Frequent hyperglycemia, on the other hand, can lead to increased HbA1c, and a subsequent increase in the risk of retinopathy and nephropathy [134,135], neuropathy [136], and all-cause morbidity and mortality from CVD and CHD [137]. Finally, greater glycemic variability can result in more endothelial dysfunction [138,139], increased oxidative stress, inflammation, and higher bone fragility [22,140,141], and classic CVD risk markers [142]. These adverse health outcomes underscore the importance of developing specific interventions and safety recommendations for specific populations.

### Can Resistance Exercise Be the Answer to Healthy Aging in Post-Menopausal Women with T1D?

As nothing is known about blood glucose changes during exercise in post-menopausal women with T1D, there are no recommendations on how to maintain safe glucose levels in this high-risk population. In addition, as a result of improvements in diabetes care, there are more women with T1D reaching menopause, and living for many years post menopause than ever before. Ensuring equitable access to, and maximal benefits from exercise and physical activity in this population is therefore of high importance.

In this regard, resistance exercise may be a promising preventive therapy to maintain health and mobility, along with preventing frailty in this particular population. As discussed, resistance exercise causes improvements in muscle strength [143], muscle quality [11], and bone density [144]. It has also been argued that resistance training may be as effective or superior to other forms of exercise with respect to treating co-morbidities associated with CVD such as sarcopenia, impaired glucose handling, and lipid metabolism [145]. Enhanced vascular condition, reduced resting blood pressure, improved body composition and mobilization of visceral and subcutaneous abdominal fat are among other cardiovascular benefits of resistance training [146]. It is therefore reasonable to consider resistance exercise as suitable for primary and secondary prevention of CVD [145]. In addition, resistance training has been associated with mental health benefits such as improvements in self-rated quality of life [147].

Although individuals with T1D may experience the same health benefits of resistance training as those without T1D, studies are extremely limited regarding the effects of resistance exercise with respect to glycemic variability in this population. While it is reported that aerobic exercise can increase the risk of hypoglycemia in T1D during activity, acute exercise studies indicate that anaerobic forms of exercise may reduce this risk [148]. Studies of resistance exercise in T1D showed that it is associated with greater blood glucose stability [149], and a lower risk of hypoglycemia during exercise [15] compared to aerobic exercise. A study comparing the acute glycemic effects of resistance exercise (3 sets of 7 exercises at 8-RM) and aerobic exercise (45 min of running at 60% of VO_2_ peak) in physically fit individuals (mean age 31.8 ± 15.3 years) showed that plasma glucose decreased rapidly during aerobic exercise, while resistance exercise caused less initial decline in blood glucose during the activity [15]. In addition, performing resistance exercise (3 sets of 7 exercises at 8-RM) prior to aerobic exercise (45 min of running at 60% VO_2_ peak) attenuated the decline in blood glucose associated with aerobic exercise and improved glycemic stability throughout the exercise session [149], compared to when these exercises were performed in the reverse order.

The protective effects resistance exercise may offer against hypoglycemia would be mediated in part by catecholamines [150]. Because epinephrine response to exercise tends to diminish with age [150], older women may not benefit as much as others from the protective effect of elevated catecholamines against hypoglycemia. In addition, early evidence that resistance exercise may be associated with higher rates of nocturnal hypoglycemia [15] should be taken into consideration when conducting research with this population.

The Diabetes Canada Clinical Practice Guidelines [151] and the American Diabetes Association’s position statement on exercise and physical activity in diabetes [152] both recommend the inclusion of regular resistance exercise for individuals with T1D. When combining aerobic and resistance exercise, the order in which the exercises are performed should also be taken into account as it can affect blood glucose levels in those with T1D [149]. However, as these recommendations are based on relatively young individuals with T1D, the effects of resistance exercise before or after an aerobic exercise session, or alone, in older adults with T1D and especially in post-menopausal women with T1D remain to be examined. Studies examining the acute effects of resistance exercise on blood glucose in this population are essential in order to ascertain its safety prior to implementing any type of long-term training study.

## 6. Conclusions

Despite the well-documented benefits of exercise, the full range of risks and benefits with respect to the health and wellness of post-menopausal women with T1D have yet to be studied. In addition, the majority of exercise studies were conducted on pre-menopausal women; thus, hormonal differences between pre-menopausal and post-menopausal women, which could play a significant role in fuel utilization and glucose response during exercise, have not been examined. In older adults, and in particular older women with T1D, the acute glycemic effects of exercise are unknown. Ascertaining these effects is essential to removing barriers to exercise and physical activity, such as fear of hypoglycemia and loss of control over blood glucose levels in older women with T1D. A greater understanding of the impacts of age, sex, and gender on the acute and training responses to exercise in post-menopausal women with T1D is necessary to reduce the burden of complications, prevent frailty, and improve quality of life in this high-risk population.

## Figures and Tables

**Table 1 ijerph-18-08716-t001:** Changes in musculoskeletal and cardiometabolic parameters and quality of life in response to physical activity, and aerobic and/or resistance exercise interventions in post-menopausal women.

Study	Number of Participants	Type of Physical Activity/Exercise	Intensity/Frequency	Program Duration	Outcome
STUDIES INVOLVING AEROBIC EXERCISE AND UNSPECIFIED PHYSICAL ACTIVITY					
Juppi et al. 2020 [35]	234	Habitual physical activity (observational data)	At least 150 min of moderate-to-vigorous PA/week, ≈ 21 min/day	Women were followed from peri to early post menopause	While menopausal transition decreased lean body mass and index and appendicular lean mass and index, physical activity was positively associated with maintained lean body mass (r = 0.182) and appendicular lean mass and index (r = 0.235 and r = 0.238, respectively)
Mazurek et al. 2017 [36]	35	Physical activity	2 weeks moderate-intensity physical training program (2.5–5.0 METs, 3 times/day, 40–75 min/session, at 40–60% of MHR) followed by 3 months of organized home-based physical activity targeting all major muscle groups	2 weeks and 3 months	Physical activity reduced systolic and diastolic blood pressure, and reduced BMI, waist-to-hip ratio, and LDL-C as compared to baseline (data provided as figures). Among participants with organized physical activity, 40.6% of women met the baseline criteria of metabolic syndrome. After two weeks of physical exercise, this percentage decreased to 18.7%, mainly due to the reduction in the above-mentioned risk factors
Hagner et al. 2009 [37]	168 (pre-, peri-, and post-menopausal women)	Aerobic exercise	Moderate-intensity Nordic walking program, three 90-minute sessions, average heart rate of 100–140 bpm	12 weeks	Exercise improved VO_2_ max, reduced BMI, waist circumference, and total fat mass, increased HDL-C, and deceased LDL-C, cholesterol, and triglycerides after 12 weeks in pre-, peri-, and post-menopausal women (except for HDL level in post-menopausal women)
Mason et al. 2013 [38]	117 (Exercise group) 98 (Control group)	Aerobic exercise	Moderate-to-vigorous intensity, 70–85% MHR, 45 min/day, 5 days/week	12 months	Intervention significantly preserved appendicular lean mass (%Δ: −0.12) and skeletal muscle index (%Δ: 0.4) compared to controls (%Δ: −1.2 and −1.5, respectively), despite no change in total lean mass
Mason et al. 2013 [38]	118 (Reduced-calorie diet group) 117 (Reduced-calorie diet with exercise)	Aerobic exercise	Moderate-to-vigorous intensity, 70–85% MHR, 45 min/day, 5 days/week	12 months	Exercise + diet attenuated the loss of appendicular lean mass and skeletal muscle index (%Δ: −1.4 and −1.0, respectively) as compared to the diet group (%Δ: −2.9 and −3.1, respectively)
Friedenreich et al. 2011 [39]	155 (Exercise group) 156 (Control group)	Aerobic exercise	Moderate-to-vigorous intensity, 45 min at 70–80% of HRR for at least half of the workout time, 5 times/week	12 months	Changes in all measures of adiposity were observed in exercisers relative to controls (the mean difference between groups: −1.8 kg for body weight; −2.0 kg for total body fat; −14.9 cm^2^ for intra-abdominal fat area; and −24.1 cm^2^ for subcutaneous abdominal fat area). Greater body fat losses were found with increasing volume of exercise (more than 225 min per week)
Gonzalo–Encabo et al. 2019 [40]	200 (High-dose group) 200 (Moderate-dose group)	Aerobic exercise	300 min a week (high dose) compared to 150 min a week (moderate dose) aerobic exercise	12 months 24 months	Significantly higher BMD (0.006 g/cm^2^ higher after 12 months and 0.007 g/cm^2^ higher after 24 months) in the high-dose exercise group as compared to moderate-dose exercise group
**STUDIES INCLUDING RESISTANCE EXERCISE**					
Teoman et al. 2004 [41]	41 (Exercise group) 40 (Control group)	Combined aerobic and resistance training program	Aerobic (65–70% MHR) and resistance training program 3 times a week for 6 weeks, starting at 30 min (including warm-up and cool-down) and increasing by 20 min over the training period	6 weeks	Significant improvements in all six markers of quality of life (physical mobility, pain, sleep, energy, social isolation, emotional status) in the exercise group as compared to control at the end of the 6th week of the training program
Villaverde–Gutiérrez et al. 2006 [42]	24 (Exercise group)24 (Control group)	Combined aerobic, resistance, flexibility, and relaxation exercises	2 supervised sessions of 30 to 60 min per week	12 months	The health-related quality of life significantly improved after the intervention in the exercise group (16.58 pre-exercise vs. 18.58 post-exercise), while it became significantly worse in the control group at the end of the study as compared to the beginning (11.96 vs. 14.12, respectively)
Figueroa et al. 2003 [43]	20 and 24 (Exercise groups with and without HRT, respectively) 22 and 28 (Control groups with and without HRT, respectively)	Combined resistance training and weight-bearing and non-weight-bearing aerobic exercise program	Resistance (free-weights with machines at 70–80% of 1-RM, 2 sets/day, 3 days/week). Aerobic (50–80% of MHR, 60–75 min/session, 3 days/week)	12 months	Combined exercise significantly increased total body (11.6%), arm (14.7%), and leg (11.0%) lean soft tissue mass, and decreased percentage of body fat (−22.9%), independent of HRT
Wooten et al. 2011 [44]	9 (Exercise group) 12 (Control group)	Resistance training program	10 exercises for 2 sets at 8-RM and the 3rd set to failure, 3 days/week	12 weeks	Significant reductions in total cholesterol (23.6%), LDL-C (28.5%), non-HDL-C (27.0%), and HDL_3_-C (24.1%) in the exercise group as compared to control following 12 weeks of resistance exercise
Ogwumike et al. 2011 [45]	90 (Exercise group) 85 (Control group)	Endurance exercise program	10 stations of circuit training exercises at 60–80% of HRR, 3 days/week	12 weeks	Significant reduction in the waist-to-hip ratio between baseline and end of 12th week in both peri-menopausal (0.86 ± 0.08 vs. 0.71 ± 0.07) and post-menopausal (0.88 ± 0.06 vs. 0.77 ± 0.07) exercise groups, with no significant changes in the control groups
Conceição et al. 2013 [46]	10 (Intervention group)10 (Control group)	Resistance training program	3 sets of 8−10 repetitions at 50–70% of 1-RM, 3 times/week, with a progressive weekly increase in load	16 weeks	Intervention decreased the metabolic syndrome severity Z-score (*p =* 0.0162) while lowering fasting blood glucose (−13.97%), improving lean body mass (2.46%), decreasing body fat percentage (−6.75%), and increasing muscle strength (41.29% for leg press and 27.23% for bench press) in exercisers as compared to controls
Watson et al. 2018 [47]	43 (High-impact training group) 43 (Control: low-impact training group)	Resistance training program	Supervised twice weekly HiRIT, compared to home-based low impact training of identical frequency and duration	8 months	HiRIT effects were superior to controls for lumbar spine BMD (2.9 ± 2.8% vs. –1.2 ± 2.8%), femoral neck BMD (0.3 ± 2.6% vs. –1.9 ± 2.6%), femoral neck cortical thickness (13.6 ± 16.6% vs. 6.3 ± 16.6%), height (0.2 ± 0.5 cm vs. –0.2 ± 0.5 cm), and all functional performance measures (*p* < 0.001)
Gómez–Tomás, et al. 2018 [48]	18 (Intervention group) 20 (Control group)	Resistance training program	6 exercises for whole-body training involving major muscle groups, 3 sets of 10 repetitions, 3 days/week	12 months	Exercise decreased weight (1.31 ± 1.49 kg decrease), waist circumference (2.67 ± 2.61 cm decrease), total cholesterol (15.72 ± 46.47 mg/dL decrease), LDL-C (16.77 ± 41.74 mg/dL decrease), and C-reactive protein (0.81 ± 1.78 mg/L decrease). No significant difference was found in HDL-C or triglycerides
Bea et al. 2010 [49]	65 (Exercise group) 32 (Crossovers) 25 (Sedentary controls)	Resistance training program	Supervised 8 exercises targeting major muscle groups, 2 sets of 8 repetitions at 70–80% of 1-RM, 3 times/week, plus progressive weight-bearing activity	6 years	Weight gain occurred in a stepwise fashion over the 6 years with controls gaining the greatest amount of weight (2.1 ± 4.3 kg controls, 0.7 ± 4.4 kg crossovers, 0.4 ± 6.2 kg exercisers). Similar to weight, gain in total body fat was also significant between baseline and 6 years in controls only (1.9 ± 4 for controls, 0.4 ± 3 for crossovers, and 0.3 ± 6 for exercisers)

PA: physical activity; METs: metabolic equivalents; MHR: maximal heart rate; bpm: beats per minute; HRR: heart rate reserve; 1-RM: one-repetition maximum; HRT: hormone replacement therapy; HiRIT: high-intensity resistance and impact training; BMI: body mass index; BMD: bone mineral density; VO_2_ max: maximal oxygen consumption; HDL-C: high-density lipoprotein cholesterol; HDL_3_-C: high-density lipoprotein 3 cholesterol; LDL-C: low density lipoprotein cholesterol.

## Data Availability

Not applicable.

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
