# Peer review of "Can Resistance Exercise Be a Tool for Healthy Aging in Post-Menopausal Women with Type 1 Diabetes?"

_ijerph, 2021, doi:10.3390/ijerph18168716_

Round 1
Reviewer 1 Report
The paper is well referenced and reviews the literature concerning exercise, postmenopausal women, and Type 1 diabetes extensively.
Therefore, it deserves publication beyond any doubt.
However, the following comments might help further improve the manuscript:
- line 57: please expand on another complex possible metabolic mechanism underlying bone fragility in type 1 diabetes: i.e., hyperglycemia, oxidative stress, and the accumulation of advanced glycation endproducts compromising collagen properties, increasing marrow adiposity, and releasing inflammatory factors and adipokines from visceral fat are known to alter osteocytes function.
- line 93: for the sake of inexperienced readers, please explain what skeletal muscle index is.
- line 133: here a short comment on the abovementioned metabolic bone damage seems appropriate too.
- line 150: for the sake of inexperienced readers, please explain what the metabolic syndrome severity Z-score is.
- lines 290-298: can the Authors exclude a two-way mechanism? in other words, chronic disabling complications might hinder physical activity as well.....
- line 413: here, again, I would add something on bone fragility depending on severe glucose oscillations.
Author Response
Please see our responses in the attached document

Reviewer 2 Report
Dear Authors,
I enjoyed reading the review you provided. The topic is relevant and current knowledge on the types of exercise, especially strength, on BMD of T1 postmenopausal women is clearly presented.
The article is well organized and relevant literature is cited. There are no many RCTs confirming the type and intensity of resistance trainings in the specific population of T1DM postmenopausal women and the authors refer primarily to the data available from overall population of postmenopausal woman. Moreover, they comment on the potential benefits of exercise on cardiovascular and metabolic health of postmenopausal women, which are a concern due to higher risk of CV death and morbidity in the mentioned population.
Reviewer 3 Report
This review highlighted the effect of resistance exercise on management of physical and mental health in post-menopausal women with type 1 diabetes. This is interesting to me; however, some points are needed to revise.
Given below are the comments regarding the paper:
1. I recommend you to change some headings’ numbers as follows: “2.1” to “3,” “2.1.1” to “”3.1,” “2.1.2” to “3.2,” “2.1.3” to “3.3,” “3” to “4,” “3.1” to “4.1,” “3.2” to “4.2,” “3.2.1” to “4.2.1,” “3.2.2” to “4.2.2,” “3.3” to “5,” “3.3.1” to “5.1,” and “4” to “6.” Exercise in post-menopausal women with T1D (“3.3.”) is main theme in this paper, thus, I recommend separating that from “3. Type 1 Diabetes.”
2. There are two description of “Women” and “Females.” Please describe either one.
3. Introduction: you should describe concisely the reason why you highlight resistance exercise in this review.
4. p8-p10 lines 257-383: I think that these paragraphs are slightly verbose. The last sentence before them (p7 lines 253-256) have shown that it is necessary to focus on physical activity and exercise for the management of menopausal symptoms in women with T1D. I think that this sentence leads to the “3.3. Exercise in Post-Menopausal Women with T1D,” hence, a more concise description would be preferable in these paragraphs.
Round 2
Reviewer 3 Report
The authors revised the last manuscript well. I will accept this.